# Engineered Josephson diode effect
# in kinked Rashba nanochannels

Alfonso Maiellaro[1,2], Mattia Trama[1,3], Jacopo Settino[4,5],
Claudio Guarcello[1,2], Francesco Romeo[1,2] and Roberta Citro[1,2]

**1** Dipartimento di Fisica "E.R. Caianiello", Università di Salerno,
Via Giovanni Paolo II, 132, I-84084 Fisciano (SA), Italy
**2** INFN, Sezione di Napoli, Gruppo collegato di Salerno, Italy
**3** Institute for Theoretical Solid State Physics, IFW Dresden,
Helmholtzstr. 20, 01069 Dresden, Germany
**4** Dipartimento di Fisica, Università della Calabria,
Via P. Bucci Arcavacata di Rende (CS), Italy
**5** INFN, Gruppo collegato di Cosenza, Italy

## Abstract

The superconducting diode effect, reminiscent of the unidirectional charge transport in semiconductor diodes, is characterized by a nonreciprocal, dissipationless flow of Cooper pairs. This remarkable phenomenon arises from the interplay between symmetry constraints and the inherent quantum behavior of superconductors. Here, we explore the geometric control of the diode effect in a kinked nanostrip Josephson junction based on a two-dimensional electron gas (2DEGs) with Rashba spin-orbit interaction. We provide a comprehensive analysis of the diode effect as a function of the kink angle and the out-of-plane magnetic field. Our analysis reveals a rich phase diagram, showcasing a geometry and field-controlled diode effect. The phase diagram also reveals the presence of an anomalous Josephson effect related to the emergence of trivial zero-energy Andreev bound states, which can evolve into Majorana bound states. Our findings indicate that the exceptional synergy between geometric control of the diode effect and topological phases can be effectively leveraged to design and optimize superconducting devices with tailored transport properties.

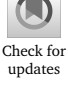

# 1  Introduction

A superconducting diode effect (SDE) refers to a phenomenon observed in certain superconducting devices where the flow of electrical current is significantly influenced by the direction of the current biasing the junction [1]. It essentially enables supercurrent to flow preferentially in one direction, providing new functionalities for superconducting circuits, such as superconducting quantum interference devices (SQUIDs), superconducting magnetic energy storage systems, and superconducting digital electronics. In recent years, there has been a significant progress towards the realization of the diode effect in superconducting heterostructures [2–14]. This phenomenon has also been investigated in several Josephson junction (JJ) systems, reflecting the advancement from pure theoretical studies [15–23] to tangible implementations. The latters exploit different configurations, and extend to the inclusion of unconventional superconductors [24–29], graphene [30,31], and topological materials [32–36]. The effectiveness of these proposals is typically evaluated through the SDE efficiency, which is determined by the difference of the Josephson critical currents when the junction is biased in opposite directions normalized to the sum of these currents. It has been explicitly demonstrated how nonreciprocity of supercurrent is intertwined with underlying symmetries of the system. In fact, nonreciprocity driven by breaking of space-inversion symmetry and time reversal symmetry, in the presence, e.g., of Rashba spin-orbit interaction (RSOC) and an applied magnetic field, has been widely investigated. In particular, evidences of nonreciprocity have been recently observed in the magnetic field dependence of the critical current in JJs at oxides interfaces, which are known to behave as Rashba-like systems [37,38]. An asymmetry in the critical current as a function of the applied magnetic field, $I_c(H) \neq I_c(-H)$, has been revealed, alongside an anomalous amplification of the critical current attributed to the emergence of Majorana bound states (MBSs) [39,40].

On the other side, the interplay between RSOC and geometric deformation of the JJ can also generate intriguing effects on the Josephson current, inducing an anomalous phase shift in the ground state of the junction [41, 42] and favouring the nonreciprocity of supercurrent [43–46]. The latter originates from non-abelian phases controlled by the RSOC [47], impacting on the overall transport properties of the system. Geometric control of Josephson coupling has been also recently demonstrated in twisted cuprate van der Waals heterostructures [48–50] and the emergence of SDE in artificial junctions composed of twisted layers of $Bi_2Sr_2CaCu2O_{8+\delta}$ [51,52] has been shown. These evidences suggest that a rich phenomenology could emerge for the non-reciprocal transport of supercurrent in the presence of RSOC, modulated by the geometric deformation of the junction and in the vicinity of topological phases. Understanding and harnessing this interplay hold great promise for the development of superconducting diode devices with enhanced functionalities, ranging from efficient energy conversion to robust quantum information processing.

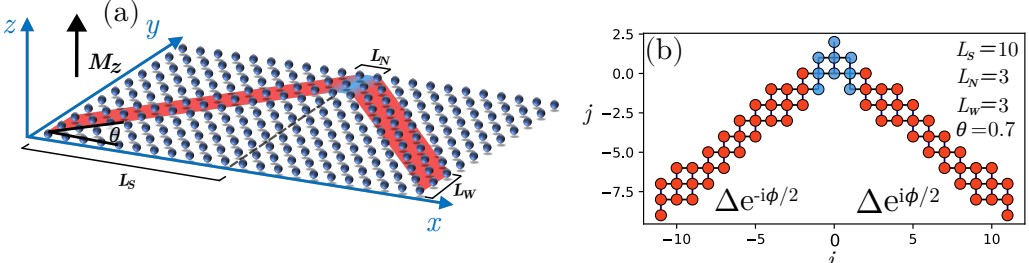

Figure 1: (a) Cartoon of the system. (b) Kwant [53] discretization $(x, y) = (ai, aj)$ with $a = 1$ for a set of representative parameters. The junction is subject to an out-of-plane magnetic field with Zeeman energy $M_z$, to the geometrical offset angle $\theta$ and to the superconducting phase difference $\phi$. The red and blue colours refer to the superconducting and normal regions, respectively.

To this aim, we analyze the diode effect in a short superconductor-normal-superconductor (SNS) junction with a kinked nanostrip geometry in the presence of strong Rashba spin-orbit coupling (RSOC). The kink angle of the Josephson junction affects transport properties by modifying the energy profile and altering the conductance. It influences the critical current and introduces local variations in hopping connectivity. We show a geometrically controlled SDE by changing the kink angle $\theta$ and map the diode efficiency by scanning the angle and the external magnetic field perpendicular to the RSOC plane. We demonstarte that the breaking of time-reversal and inversion symmetry are crucial ingredients for the manifestation of the SDE. Interestingly the maximal efficiency is obtained on the verge of the topological phase where non-trivial zero-energy bound states emerge. Beyond the non-reciprocal behavior of the supercurrent, we demonstrate the evolution of Andreev bound states at the Fermi level from trivial to non-trivial Majorana bound states (MBSs) with unique spectral characteristics. The organization of the paper is as follows: In Sec. 2, we introduce the model and formalism. In Sec. 3, we discuss the rich behavior of the supercurrent as a function of the kink angle and applied magnetic field, focusing on the supercurrent diode effect (SDE). Finally, in Sec. 4, we present our conclusions and provide an outlook.

## 2 Model and formalism

We consider a kinked nanostrip superconducting junction on a two-dimensional gas with Rashba interaction and variable kink angle. The superconducting leads are considered at a relative angle $\theta$ with respect to the positive orientation of the $x$-axis of the square lattice (see Fig. 1). The region defined by the junction determines the system domain $\mathcal{D}$, with its area given by $A = L_W(2L_S + L_N)$. The lengths of the superconducting and normal regions projected along the $x$-axis are denoted by $L_S$ and $L_N$ respectively, while $L_W$ represents the width of the junction (see supplemental materials at A). For every kinked realization $\theta$ the system experiences random surface roughness where some neighboring sites are connected by hopping and others are not (see Fig. 1(b)). The latter effects simulate the detrimental impact of lattice disorder, which is extremely relevant in experiments.

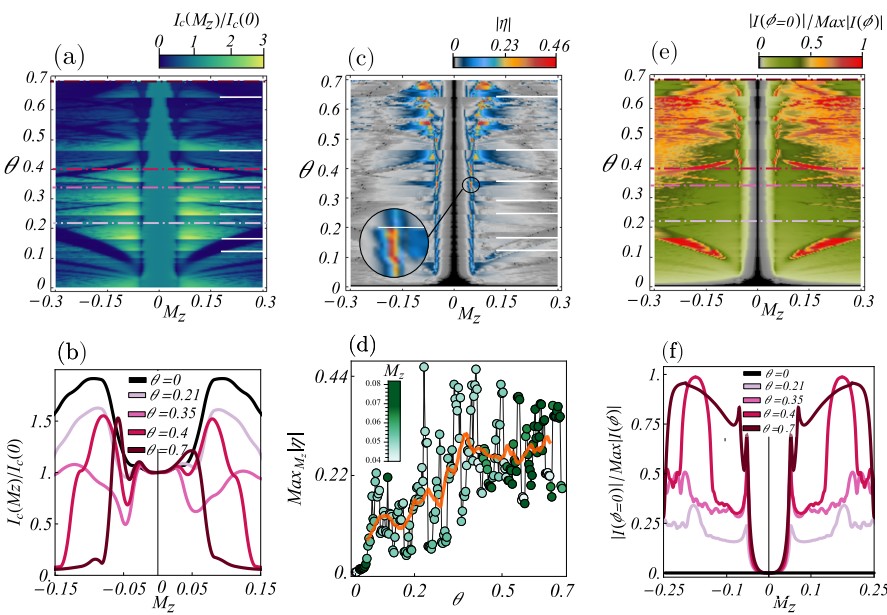

Figure 2: (a) Phase diagram of the critical current as a function of $\theta$ and $M_z$, the white lines correspond to $\theta_n = \arctan(3/n)$ with $n = 4, 6, 8, 10, 12, 18, 24$. (b) Selected horizontal cuts from panel (a) showing a diode effect strongly marked for high values of $\theta$. (c) Phase diagram of the diode efficiency $\eta$ and (d) field-optimization of $\eta$ as a function of $\theta$. The superimposed orange line depicts the moving average computed over a 25-point data window. (e) Phase diagram of the zero-phase current behavior, with select horizontal cuts provided in panel (f). Notably, red regions correspond to the nucleation of $\pi/2$-junctions, easily recognizable by the enhanced regions in the magenta and dark magenta curves of the panel (f) ($\theta = 0.4, 0.7$). The $\theta$ values chosen in panels (b) and (f) are also signaled in the phase diagrams (a) and (e) by means of horizontal dashed lines.

In the absence of superconductivity, the Hamiltonian can be described as $H = H_0 + H_{SO} + H_M$, with:

$$H_0 = \sum_{i,j}\Big[(-\mu + 4t)\Psi_{i,j}^\dagger \sigma_0 \Psi_{i,j} - t\left(\Psi_{i,j}^\dagger \sigma_0 \Psi_{i+1,j} + \Psi_{i,j}^\dagger \sigma_0 \Psi_{i,j+1} + H.c.\right)\Big], \tag{1}$$

$$H_{SO} = i\alpha\Big(\sum_{i,j}\Psi_{i,j}^\dagger \sigma_y \Psi_{i+1,j} - \Psi_{i,j}^\dagger \sigma_x \Psi_{i,j+1}\Big) + H.c., \tag{2}$$

$$H_M = \sum_{i,j}\Psi_{i,j}^\dagger M_z \sigma_z \Psi_{i,j}, \tag{3}$$

where $\Psi_{ij} = (c_{ij,\uparrow}\ c_{ij,\downarrow})^T$ denotes a vector whose components are the electron annihilation operators for a given spin and position. The terms $H^0$, $H^{SO}$, and $H^M$ in the Hamiltonian represent the kinetic energy, spin-orbit coupling, and Zeeman interaction, respectively. The summation over indices $(i, j)$ spans the system domain $\mathcal{D}$. $\sigma_i$ are the Pauli matrices, while $\sigma_0$ indicates the identity matrix. In order to introduce superconducting correlations, we add to the Hamiltonian $H$ the mean-field pairing contribution:

$$H_P = \sum_{i,j\in S}\Delta_{i,j}\ c_{i,j,\uparrow}^\dagger c_{i,j,\downarrow}^\dagger + H.c. \tag{4}$$

The singlet order parameter $\Delta_{i,j} = \Delta\ e^{i\varphi(i,j)}$ is present exclusively within the two superconducting leads, so that $\varphi(i,j)$ takes the values $\varphi_L$ or $\varphi_R$ when $(i,j)$ belong to the left or right lead, respectively, with $\phi = \varphi_L - \varphi_R$ representing the phase gradient (Fig. 1(b)).

The Josephson current of a short SNS junction with translational invariant leads can be efficiently calculated by using the subgap Bogoliubov-de Gennes (BdG) spectrum of a system with truncated superconducting leads as $I(\phi) = -(e/\hbar) \sum_n dE_n/d\phi$ [54], $E_n$ being the subgap energies of the BdG spectrum. The finite lead approach is valid as long as the short junction limit is considered, implying that the dimension $L_N$ of the normal part of the SNS junction is smaller or comparable with the BCS coherence length $\xi$ (i.e., $L_N \ll \xi$). Once $I(\phi)$ has been obtained by exact diagonalization procedure of the BdG Hamiltonian, the maximum (absolute value of minimum) of $I(\phi)$ yields the critical current in the positive (negative) direction. Due to the non-Abelian phases arising from the combined action of RSOC and the kinked geometry, currents flowing in junctions characterized by angles $\theta$ and $-\theta$ exhibit opposite signs, implying the symmetry rule $I(\theta, M_z, \phi) = -I(-\theta, M_z, -\phi)$. Furthermore, reversing the direction of the Zeeman term results in a reversal of current sign, so that $I(\theta, M_z, \phi) = -I(\theta, -M_z, -\phi)$. This symmetry allows for the determination of the critical current in either positive or negative directions by simply exchanging the sign of $M_z$.

The Hamiltonian is numerically treated using Kwant [53] and solved with the help of NUMPY routines [55]. We explore a wide range of angles ($\theta$), varying the Zeeman energy ($M_z$), and the phase difference ($\phi$). Numerical simulations are performed in units of the tight-binding hopping integral ($t = 1$). The other parameters are fixed at $\Delta = 0.05$ and $\alpha = 0.01$, the latter value being typical for oxide interface RSOC [56–62]. Throughout the paper, we also set $L_S = 100$, $L_N = 3$, and $L_W = 3$. The chemical potential is expressed as $\mu = \mu_0 + e_0$, where $e_0$, determining the filling, represents the energy offset measured from the bottom ($\mu_0$) of the Rashba lowest energy band. The parameter $\mu_0$ is consistently computed across all realizations of $\theta$.

## 3 Results

In Fig. 2(a), we present the critical current $I_c$ as a function of the magnetic energy $M_z$ and the junction angle $\theta$. We observe that for $M_Z \lesssim 0.05$, the critical current remains nearly unaffected by variations in $\theta$, so that $I_c \approx I_c(0)$. However, for higher $M_z$ values, an anomalous enhancement of the critical current occurs, as indicated by the yellow regions in the figure. This enhancement has been previously documented in Rashba-like materials and can be attributed to a topological phase with localized MBSs pinned at the junction ends [39]. The field $M_z \sim 0.05$ represents the critical threshold for the topological phase transition (details in Supplemental material A). Interestingly, the enhancement phenomenon is not merely affected by the applied magnetic field but also critically depends on the junction angle. Specifically, $I_c$ shows larger values at angles $\theta \gtrsim \theta_n$, indicated by white lines in the figure, of the form $\theta_n = \arctan(3/n)$, with $n$ being even integers. These angles define the condition for which translational invariance is restored in each arm of the junction (see Supplemental material A). Moreover, the complex influence of $M_Z$ and $\theta$ on the critical current also induces dark regions within the $I_c(\theta, M_Z)$ map, where the critical current is drastically suppressed. The suppression of the critical current documented above is related to the interference of a few transverse modes with distinct scattering phases. Under appropriate conditions, transverse modes physics induces Fraunhofer-like patterns in the $I_c$ vs $M_z$ curves, highlighting the interference-related nature of the observed phenomenon. Interestingly, the scattering phases mentioned are controlled by the kink angle and the external magnetic field, thus providing important experimental knobs to control the distinct transport regimes of the junction.

The curves in Fig. 2(b) correspond to the horizontal cuts shown in Fig. 2(a) and evidence a strong asymmetry of the $I_c$ vs $M_z$ curves when positive and negative magnetic fields values are considered. In order to quantify the magnitude of this asymmetry, we compute the SDE efficiency [1]

$$\eta(\theta, M_z) = \frac{I_c(\theta, M_z) - I_c(\theta, -M_z)}{I_c(\theta, M_z) + I_c(\theta, -M_z)}. \tag{5}$$

The efficiency $\eta$, presented in Fig. 2(c), exhibits a pronounced peak in the vicinity of the topological transition ($M_z \sim 0.05$) for small values of $\theta$ ($\lesssim 0.4$). Intriguingly, efficiency also exhibits a notable angular dependence: the recovery of translational symmetry inhibits the SDE (see zoomed area of Fig. 2(c)), making it more pronounced for angles just below these $\theta_n$. For $\theta \gtrsim 0.4$, the behavior of $\eta$ becomes more complex. The peaks of efficiency migrate away from the topological transition region in an irregular fashion. In Fig. 2(d) we plot the maximal diode efficiency at fixed $\theta$ among the values of $M_z$ considered. In this case, when optimizing with respect to the magnetic field, we find that $\eta$ fluctuates with angle, reaching peaks of $\sim 0.46$ at certain angular and magnetic coordinates, following an increasing trend, which is marked by an orange line. The magnetic fields that maximize $\eta$, reported as a green color gradient, increase on average with $\theta$. The moving average of the $\theta$ dependence of the maximum efficiency, indicated by the orange line in Fig. 2(d), provides predictions about the evolution of the diode efficiency in kinked nanojunctions, as realizable in experiments. Indeed, from the experimental side, resolution limits of the fabrication process may prevent achieving the perfect alignment associated with a fixed value of $\theta$. Thus, Fig. 2(d) presents a crucial figure of merit for a geometry-controlled nanodevice exhibiting SDE.

Beyond the investigation of the diode effect, signatures of the so-called *anomalous Josephson effect* also emerge. Specifically, we are dealing here with a particular kind of unconventional junction, i.e., the $\phi_0$-*junction* [63–70], characterized by a non-trivial ground phase ($\phi_0 \neq 0$ or $\pi$) and a significant cosine component in the CPR. This gives rise to a non-zero current at $\phi = 0$, i.e., the *anomalous Josephson current*. Thus, a $\phi_0$-junction can yield a constant phase bias $\phi = -\phi_0$ in an open-circuit configuration, or even a current $I \propto \sin(\phi_0)$ if inserted into a closed superconducting loop. The crossover from a conventional to an anomalous Josephson system, at $\theta \neq 0$ and $M_z \neq 0$, is depicted in Fig. 2(e); here, we focus on the anomalous Josephson current, i.e., $I(\phi = 0)$, normalized to the maximum current value. We notice that, as long as $M_z \leq 0.05$, $I(\phi = 0)$ remains small and insensitive to $\theta$. Furthermore, the points where $I(\phi = 0)$ is maximized correspond to regions with suppressed critical current in Fig. 2(a) (dark areas of the density plot). To better visualize the anomalous Josephson effect, in Fig. 2(f) we collect a few selected slices from Fig. 2(e), taken at the same angles as in Fig. 2(b). From small (pink) to high (bordeaux) angles, $I(\phi = 0)$ increases, also approaching conditions (corresponding to the red regions in Fig. 2(e)) at which the CPR takes on a cosine-like profile, implying the formation of a $\pi/2$-junction with $I(\phi = 0) \approx Max_\phi(I(\phi))$.

Apart from MBSs, the peculiar intertwining between the effects controlled by the geometry (kink angle) and the magnetic energy engenders additional sub-gap quasiparticle modes. We classify all the sub-gap states into three categories, distinguished by their spectral properties and influence on the CPR (Fig. 3). Trivial Andreev bound states (ABSs) are associated with sinusoidal CPRs (Fig. 3(a)) and their probability densities are spread over the whole system size (Fig. 3(b)), carrying a sub-gap excitation energy well above the Fermi level (Fig. 3(g)). These states can be found in the region where $I_c$ decreases with respect to $M_z$. MBSs, on the other hand, are associated with an enhancement of $I_c$ vs $M_z$ curves and exhibit a sawtooth CPR with a single nodal point in the phase-periodicity range (Fig. 3(d)). They are localized at the four edges of the junction (Fig. 3(e)) and reside at the Fermi energy (Fig. 3(f)) [71, 72]. Finally, Zero-energy Andreev bound states (ZABSs) can be found in the region where $I_c \sim I_c(M_z = 0)$, before the critical current increases. The phase-space region, defined by $(\theta, M_z)$ and host-

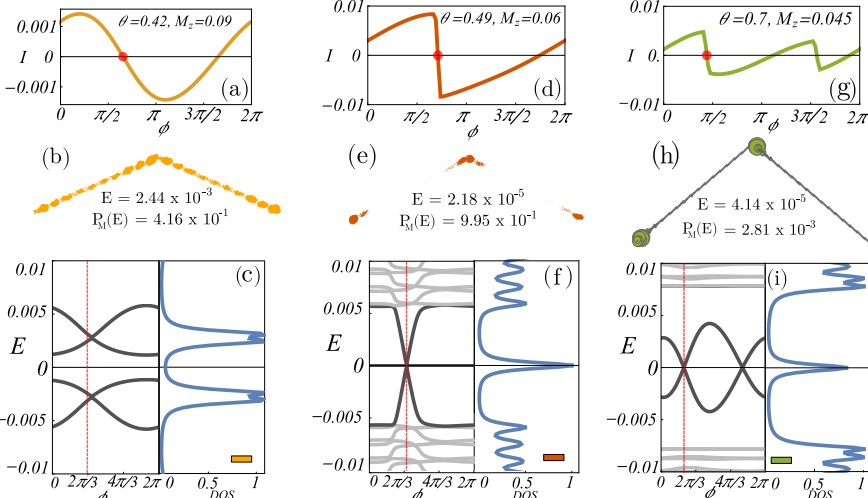

Figure 3: Classification of the sub-gap states, ABSs, MBSs and ZABSs, by means of the CPRs (a), (d), (g), the probability densities $|\psi_i|^2$ (b), (e), (h) and spectra and DOSs (c), (f), (i). The graphs and the DOSs are calculated respectively by fixing $\phi$ at the value marked by the red point in the CPRs and the vertical red cut in the spectra. The graphs reproduce the geometry of a nano-strip with $L_S = 100$, $L_N = 3$, $L_W = 3$, being the radius of the circles proportional to $|\psi_i|^2$ ($i = 1,\ldots,L$). The spectral energy $E$ and the Majorana polarization $P_M(E)$ of these states are also reported. For the current and energy plots we fix respectively $et/\hbar = 1$ and $t = 1$.

ing ZABSs, exhibits complex boundaries, the prediction of which is a rather complex task. Concerning their influence on the transport properties, ZABSs exhibit a double-sawtooth CPR (Fig. 3(g)) and, compared to MBSs, exhibit localization on a smaller number of lattice sites, placed at the left corner of the superconducting leads (Fig. 3(h)). They also share the same energy of MBSs for two specific values of the superconducting phase difference (Fig. 3(i)). However, despite their similarities to MBSs, neither ABSs nor ZABSs carry topological charge, as evidenced by the Majorana polarization ($P_M$) [73,74], which is indicated in the insets of panels (b), (e), and (f). Indeed, $P_M$ is a measure of topological quasiparticle weight in the Nambu space. The local Majorana polarization is a function of lattice sites ($i,j \in \mathcal{D}$) and energy: $P_M^{loc}(i,j,\omega) = \sum_n \left(\sum_\sigma u_{n,i,j,\sigma} v_{n,i,j,\sigma}\right)(\delta(\omega - E_n) + \delta(\omega + E_n))$, with $u$ and $v$ the particle and hole components of the Bogoliubov wavefunction, while $\sigma$ is the spin. Let us note that, by choosing $\omega = 0$, the total Majorana polarization $P_M = |\sum_{i=1}^{L_S/4} \sum_{j=1}^{L_w} P_M^{loc}(i,j,0)|$ is equal to 1 for genuine Majorana fermions while $P_M \ll 1$ for ABSs and ZABSs. Moreover, ZABs can be understood as defect states dressed by the superconducting pairing, which originates from connectivity defects or quasi-periodicity of the lattice structure [75,76]. $P_M^{loc}(i,j,0)$ defines a vector with $P_{M,x}^{loc}(i,j) = Re[P_M^{loc}(i,j,0)]$ and $P_{M,y}^{loc}(i,j) = Im[P_M^{loc},(i,j,0)]$, and both $P_{M,x}^{loc}$ and $P_{M,y}^{loc}$ are peaked functions at the system edges, i.e., they show MBSs with opposite topological charge.

## 4 Conclusions and outlooks

In summary, we have reported the emergence of the superconducting diode effect in geometry-controlled Josephson junctions based on Rashba nanostrips. By tuning parameters such as the magnetic field and the kink angle, $M_Z$ and $\theta$, respectively, we achieved control over critical current and diode efficiency, which is pivotal for device optimization. Notably, we achieved

remarkable optimization of diode efficiency, with values exceeding 40% and a minimum efficiency of 20% for $\theta \geq 0.1$. The intricate interplay between effects controlled by $\theta$ and $M_Z$ reveals additional intriguing phenomena, including the crossover from conventional to anomalous Josephson effects and the effective tuning of the quasiparticle spectrum. Our investigation identifies Andreev bound states, zero-energy Andreev bound states, and Majorana bound states as significant sub-gap excitations, each associated with distinct transport regimes of the junction. These phases are highly sensitive to the nanochannel bending, thus providing a relevant resource to optimize the SDE. Among the experimental platforms, the oxide interfaces between LAO and STO are promising candidates to realize the kinked nano-strip. Here it's possible to realize the kinked junctions with typical Rashba interaction of the order of magnitude considered in our manuscript. Moreover, different techniques are experimentally available to engineer the junctions with desired geometry, like the energy ion irradiation technique [59, 77, 78] or e-beam lithography with lift off technique [37]. Another suitable platform is represented by the flakes of $2H\text{-}NbSe_2$, a layered two-dimensional (2D) transition metal dichalcogenide, a prototypical type-II superconductor that exhibits strong anisotropic responses to external magnetic field [7]. For real systems, for which the fine geometrical tuning may not be achieved with a large enough resolution, we demonstrated the robustness of the SDE by increasing the kink angle, i.e., the higher $\theta$, the larger the SDE. Moreover, we cannot rule out the possibility that actual physical systems may exhibit asymmetries, angular misalignments, and/or irregularities at the edges. Indeed, the presence of impurities has been demonstrated to induce transition to $\pi$-junctions in spin-orbit magnetic devices [79, 80]. This aspect suggests the potential coexistence of different phases in a realistic experimental setup close to specific junction configurations, which can be further investigated. In conclusion, our findings represent a significant advancement in both the fundamental understanding and practical engineering of Rashba-based Josephson junctions for future nanoelectronic devices.

# Acknowledgments

**Funding information**   The authors acknowledge financial support from PNRR MUR project PE0000023-NQSTI-TOPQIN. M.T. and A.M. acknowledge financial support from "Fondazione Angelo Della Riccia". F.R. acknowledges funding from Ministero dell'Istruzione, dell'Università e della Ricerca (MIUR) for the PRIN project STIMO (Grant No. PRIN 2022TWZ9NR). This work is financed by the Horizon Europe EIC Pathfinder under the grant IQARO number 101115190.

# A   Supplemental material

## A.1   System parameterization

The system domain $\mathcal{D}$ of the kinked nanostrip superconducting junction can be parameterized by starting from a square lattice with integer lattice coordinates $(i, j)$. Each site, with the lattice coordinates $(i, j)$, has the real-space coordinates $(x, y) = (ai, aj)$. Specifically, the positions of the superconducting leads are defined by the following set of inequalities, $-(L_S + (L_N - 1)/2) \leq x \leq -((L_N - 1)/2 + 1)$, $\tan \theta\ x < y \leq \tan \theta\ x + L_W$, and $(L_N - 1)/2 + 1 \leq x \leq (L_S + (L_N - 1)/2)$, $-\tan \theta\ x < y \leq -\tan \theta\ x + L_W$. The normal region is positioned between the superconducting regions, defined by $-(L_N - 1)/2 \leq x \leq (L_N - 1)/2$, $-\tan \theta\ |x| \leq y \leq -\tan \theta\ |x| + L_W$. $L_N$ and $L_S$ denote respectively the lengths of the superconducting and normal regions projected along the $x$-axis and $L_W$ represents the width of the junction. The total system size is fixed as $A = L_W(2L_S + L_N)$. An illustration of the system is reported in Fig. 1 of the main text.

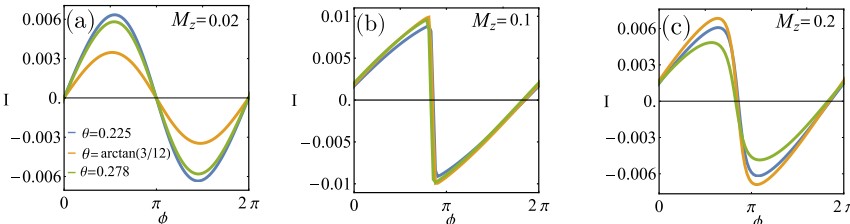

Figure 4: (a)-(c) Comparison between the CPRs of two quaiscrystals ($\theta = 0.225$, $0.278$) and of a crystal ($\theta = \arctan(3/12)$) for three selected values of $M_z$.

## A.2 Current-phase relation of crystals and quasicrystals

In Fig. 4 we present the CPRs for three values of $\theta$ and $M_z$. $\theta = \arctan(3/12)$ corresponds to one of the horizontal white lines in Fig. 2(a), while the other two angles are in its vicinity. The values of $M_z$ are selected from three distinct regimes experienced by the system: before the topological regime, Fig. 4(a), inside the topological regime, Fig. 4(b), and out of the topological regime, Fig. 4(c). We reveal that when $M_z$ surpasses the critical Zeeman threshold to enter the topological phase ($M_z \sim 0.05$) the current led by $\theta = \arctan(3/12)$ is greater than those in its vicinity. The observed enhancement can be attributed to the reconstruction of the crystalline structure with a translationally invariant unit cell ($\theta = \arctan(3/12)$).

## A.3 Further results on the anomalous Josephson effect

In Fig. 5(a) we consider a horizontal cut extracted from the phase diagram depicted in Fig. 2(b), with $\theta = 0.2$. A dip-peak structure is evident around $M_z \sim 0.055$ (vertical dashed red line), marking the critical threshold for the onset of topological phases. We observe that the spectral behavior, initially lifted from zero energy at lower fields with a sinusoidal CPR (see Fig. 5(b)), converges towards zero energy for $M_z \geq 0.055$ (see Figs. 5(c)-(d)), accompanied by the emergence of a sawtooth CPR in Fig. 5(d). The grey and black curves in the spectra represent the BdG sub-gap eigenenergies. The lowest energy states (black ones) are insensitive to the phase difference $\phi$ in the topological phase (Fig. 5(d)), the latter being a signature of Majorana character.

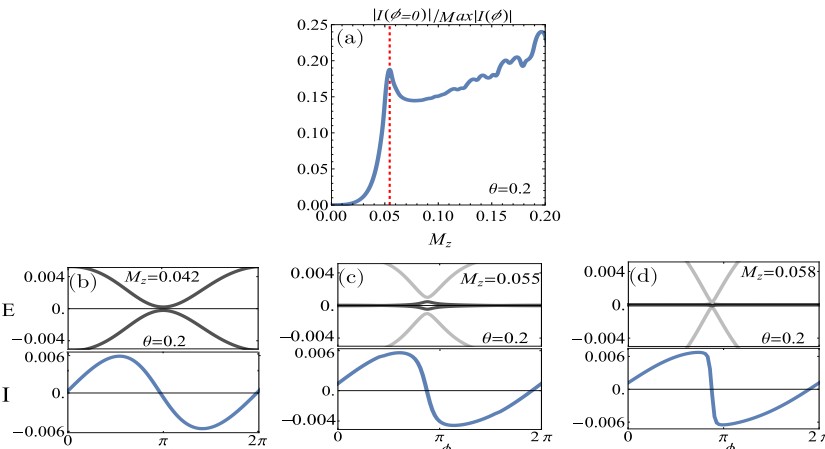

Figure 5: (a) Horizontal cut for $\theta = 0.2$ of Fig. 2(b) of the main text. (b)-(d) Spectra and CPRs evaluated accordingly to the parameters of panel (a) and for three selected values of $M_z$. Specifically $M_z$ in panel (c) is fixed accordingly to the dashed vertical red line of the panel (a).

## A.4 Crystalline configurations

In this section, we highlight that some angles can restore the translational invariance on one side of the junction. In fact, given the transverse length $L_W$, when the angle of the junction is $\theta_n = \arctan\left(\frac{L_W}{n}\right)$, with $n \in \mathbb{Q}$ and $n \geq 1$, it is always possible to find a repeated elementary cell and a longitudinal quasi-momentum $k$. $\theta_1$ represents an upper bound for the realization of the strip since above this limit there are no hopping in the $x$ direction. In Fig. 6 we highlight three examples of configurations for $L_W = 2$.

As we saw in the main text, some of these angles are very close to the enhanced diode-effect configurations. This observation raises interesting questions about the impact of delocalized states on the phase of superconducting junctions, which however are beyond the scope of this work.

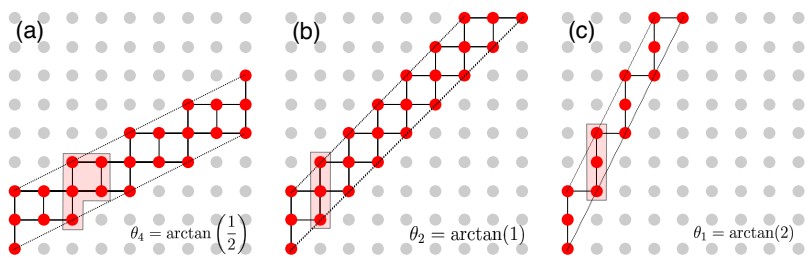

Figure 6: Crystalline realization for some benchmark angles (a) $\theta_4 = \arctan\left(\frac{1}{2}\right)$, (b) $\theta_2 = \arctan(1)$ and (c) $\theta_1 = \arctan(2)$.

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
