# Peer review of "Engineered Josephson diode effect in kinked Rashba nanochannels"

_SciPost Physics, doi:SciPost Phys. 17, 101 (2024)_

## Round 2 · Referee Report · Anonymous (Referee 6) · 2024-8-14

Report

I appreciate the authors' thorough efforts in addressing the majority of the concerns raised in my report. The clarifications provided are satisfactory, and the revisions made to the manuscript have significantly strengthened it. Specifically, the new paragraphs on disorder and on the experimental realization are valuable in highlighting the novelty of this work compared to previous studies on similar devices, and they have dissipated most of my initial skepticism about its suitability for publication. Therefore, I now support the publication of this manuscript in SciPost.

Recommendation

Publish (meets expectations and criteria for this Journal)

---

## Round 2 · Referee Report · Anonymous (Referee 2) · 2024-8-21

Report

The revised version of the manuscript now fits the standard of SciPost Physics.

Recommendation

Publish (easily meets expectations and criteria for this Journal; among top 50%)

---

## Round 2 · Author Response

Answer to Editor Dear Editor, We have thoroughly considered the referees' comments and criticisms and have modified the original manuscript accordingly, strengthening our conclusions. Enclosed, you will find a detailed response addressing each of the criticisms, along with a comprehensive list of the changes made. Referees 1, 2, 3, 4, and 5 have expressed positive assessments of our results. They have highlighted the significance of our findings and the clarity of our presentation. Their feedback indicates a strong endorsement for publication. Referee 6 has raised some concerns, which we believe are resolved by our responses and by the revised manuscript. For these reasons, we feel that our manuscript, in its present form, should be suitable for publication in SciPost.

Kind regards, on behalf of the authors, Alfonso Maiellaro

                                                                                                                             Report 1

We thank the referee for their careful reading of the manuscript and for the positive assessment of our results. We also appreciate the referee for highlighting some critical points requiring clarification. Below, we provide a detailed response to the referee’s comments and critiques:

1) The transport properties of a kinked nanostrip, influenced by Rashba spin-orbit coupling, are impacted by non-Abelian phases in the electron motion, which are tied to the specific sequence of paths followed by the current. The non-Abelian phase accumulated along each path depends on its orientation. To elucidate, let us assume a non-Abelian phase A ̂ is accumulated along the first branch, and a second phase B ̂ is accumulated along the second branch. The total non-Abelian phase A ̂B ̂ affects the quantum transport for a specific direction. Conversely, for the reversed path, the phase B ̂A ̂ is accumulated. Since A ̂B ̂≠B ̂A ̂, this results in non-reciprocal transport properties of the junction. Furthermore, due to the symmetry properties of the model, reversing the direction of the Zeeman term results in a reversal of the current sign.

2) We appreciate the referee comment about the bibliography. We will improve the references list to provide the reader with more context. Report 2 We thank the referee for their careful reading of the manuscript and for the positive assessment of our results. We also appreciate the referee for highlighting some critical points requiring clarification. Below, we provide a detailed response to the referee’s comments and critiques: 1) The coherence length can be derived by the model parameters as \xi=\hbar^2/(a m^\ast \Delta) where a, m^\ast and \Delta are respectively the lattice constant, the effective mass and the superconducting gap. Accordingly, we can estimate a length of 20 a, the latter being much smaller than the electrode length.

2) The transport properties of a kinked nanostrip, influenced by Rashba spin-orbit coupling, are impacted by non-Abelian phases in the electron motion, which are tied to the specific sequence of paths followed by the current. The non-Abelian phase accumulated along each path depends on its orientation. To elucidate, let us assume a non-Abelian phase A ̂ is accumulated along the first branch, and a second phase B ̂ is accumulated along the second branch. The total non-Abelian phase A ̂B ̂ affects the quantum transport for a specific direction. Conversely, for the reversed path, the phase B ̂A ̂ is accumulated. Since A ̂B ̂≠B ̂A ̂, this results in non-reciprocal transport properties of the junction. Furthermore, due to the symmetry properties of the model, reversing the direction of the Zeeman term results in a reversal of the current sign. Report 3 We thank the referee for their careful reading of the manuscript and for the positive assessment of our results. We also appreciate the referee for signalling out the missing references. We will add them to a new version of the manuscript. With this changes we hope that the manuscript can be accepted for publication.

1)For every θ and M_z values, we numerically identify μ_0 as the bottom of the lowest energy band in the normal phase. The energy offset measured from μ_0 is indicated as e_0. The diode is induced by both θ and M_z, as it is explained at point 2. The anomalous Josephson effect only requires the presence of M_z as discussed in ref. [39] of the manuscript.

2)We acknowledge the referee for signalling out the typo. We will substitute “green curves” with “magenta and dark magenta curves” in the manuscript.

3)According to the referee suggestions, we will add to the manuscript a short paragraph to provide the reader with more context about Majorana polarization, which has been defined in ref.s [67, 68] of the manuscript.

4) Both grey and black curves represent the BdG sub-gap eigenenergies. The black ones correspond to the lowest energy states that in the topological regime are insensitive to the phase difference ϕ, the latter being a signature of Majorana character. A sentence will be added to App C to elucidate this property. Report 4 We thank the referee for their careful reading of the manuscript and for the positive assessment of our results. We also appreciate the referee for highlighting some critical points requiring clarification. Below, we provide a detailed response to the referee’s comments and critiques:

1) The transport properties of a kinked nanostrip, influenced by Rashba spin-orbit coupling, are impacted by non-Abelian phases in the electron motion, which are tied to the specific sequence of paths followed by the current. The non-Abelian phase accumulated along each path depends on its orientation. To elucidate, let us assume a non-Abelian phase A ̂ is accumulated along the first branch, and a second phase B ̂ is accumulated along the second branch. The total non-Abelian phase A ̂B ̂ affects the quantum transport for a specific direction. Conversely, for the reversed path, the phase B ̂A ̂ is accumulated. Since A ̂B ̂≠B ̂A ̂, this results in non-reciprocal transport properties of the junction. Furthermore, due to the symmetry properties of the model, reversing the direction of the Zeeman term results in a reversal of the current sign.

2) We acknowledge the referee for signalling out these missing references. We will add to the bibliography of the manuscript.

3) Our choice of M_z is motivated by the fact that the junction is not aligned along a single direction. So that, in order to obtain a homogeneous effect of the magnetic field on the device, we choose an out of plane configuration.

4) We acknowledge the referee for signalling this weak point. We will add the mentioned lines to Fig.2 (b) and (f).

5) According to the referee suggestions, we will provide a new version of the figure with updated caption and labels.

6) According to the referee suggestions, we will add to the manuscript a short paragraph to provide the reader with more context about Majorana polarization, which has been defined in ref.s [67, 68] of the manuscript.

                                                                                                                     Report 5

We thank the referee for their careful reading of the manuscript and for the positive assessment of our results. Among the experimental platforms, we repute that the oxides interfaces between LAO and STO are suitable candidates. Here it’s possible to realize the kinked junctions with typical Rashba interaction of the order of magnitude considered in our manuscript (see Refs. [55-61]). Moreover, different techniques are experimentally available to engineer the junctions with desired geometry, like the energy ion irradiation technique (see Ref. 38 and.https://doi.org/10.3390/nano11020398; https://doi.org/10.1103/PhysRevB.96.020504) or e-beam lithography with lift off technique (see Ref. 37). The realistic length of the weak link can be approximatively down to 30 nm and the applied magnetic field, compatible with superconductivity, can be up to 200 mT (Ref. 38). The energy scale that we used in our manuscript is the hopping amplitude t whose benchmark values are of the order of 100 meV in oxides interfaces (see Ref. 60). Another suitable platform is represented by the flakes of 2H-NbSe2, a layered two-dimensional (2D) transition metal dichalcogenide, a prototypical type-II superconductor that exhibits strong anisotropic responses to external magnetic field (Ref. 7). Report 6 We thank the referee for his/her careful reading of the manuscript and for recognizing that our work represents a step forward in the characterization and optimization of SDE systems. Below, we provide a detailed response to the referee’s comments and critiques: 1) We partially agree with the referee that Rashba nanowires forming Josephson junctions with kinked/bent geometries have been previously considered, but we disagree with the statement that the novelty of our results is moderate. Indeed, we believe that our system is already a deviation from the pristine nanochannel case, and our study represents a significant advancement, both in the methodology and in the analysis, compared to the the previous papers. In the physical picture emerging in our system, the kink angle of the Josephson junction affects transport properties by modifying the energy profile, altering the conductance and showing a rich phenomenology of bound states. It influences the critical current and introduces local variations in hopping connectivity. The latter aspect, i.e. a bending-controlled Josephson junction transparency, is completely missing in the Phys Rev. B 103, 144520 (2021), which is based on a one-dimensional quantum wave guide approach. Furthermore, in our system, we introduce a transversal width L_w, playing a relevant role in the emerging physical picture. Indeed, for every kinked realization θ, the system experiences a surface roughness where some neighboring sites are connected by hopping and others are not (see Fig.1(b)). The latter effects simulate the detrimental effect of disorder, which is extremely relevant in experiments. The effect of random weak disorder (like oxygen vacancies) doesn’t affect our picture as long as the disorder potential, which couples to the density, is lower then the relevant energy scales in our system (hopping and superconductiong gap). For these reasons, we believe that our work properly addresses a physical condition presenting a relevant deviation from the pristine nanochannel case. We will add a sentence to the manuscript to further elucidate these aspects.

2) he breaking of time-reversal and inversion symmetry are believed to be crucial ingredients for the manifestation of the SDE effect. When translational invariance along the transport direction is broken (θ≠θ_n) under a magnetic field M_z, both the requirements are satisfied and a relevant SDE is observed. Conversely when the inversion symmetry along the transport direction is preserved SDE is not expected. A sentence will be added to the manuscript in order to clarify this aspect.

3) Among the experimental platforms, we repute that the oxides interfaces between LAO and STO are suitable candidates. Here it’s possible to realize the kinked junctions with typical Rashba interaction of the order of magnitude considered in our manuscript (see Refs. [55-61]). Moreover, different techniques are experimentally available to engineer the junctions with desired geometry, like the energy ion irradiation technique (see Ref. 38 and.https://doi.org/10.3390/nano11020398; https://doi.org/10.1103/PhysRevB.96.020504) or e-beam lithography with lift off technique (see Ref. 37). Another suitable platform is represented by the flakes of 2H-NbSe2, a layered two-dimensional (2D) transition metal dichalcogenide, a prototypical type-II superconductor that exhibits strong anisotropic responses to external magnetic field (Ref. 7). We will also add a paragraph to the manuscript to discuss these experimental platforms, meeting also the requirement of Referee 5.

                                                                            Summary of changes
1) We have added the references suggested by Referees 1, 3, and 4..

2) We have included two new lines in Fig. 2(b) and (f) corresponding to the case with θ=00

3) We have added two sentences in the Introduction and one sentence in the "Model and Formalism" section to reinforce the novelty of our manuscript.

4) We have fixed the typos in the caption of Fig. 2.

5) We have fixed the measurement units in the caption of Fig3.

6) We have added a new paragraph about Majorana Polarization at the end of the section: RESULTS.

7) We have added a new paragraph about experimental realization at the end of the section conclusions and outlooks.

8) We have added a new sentence in the Appendix C explaining in more detail the spectra depicted in Fig. 5.

---

## Round 2 · List of Changes

Summary of changes 1) We have added the references suggested by Referees 1, 3, and 4..

2) We have included two new lines in Fig. 2(b) and (f) corresponding to the case with θ=00

3) We have added two sentences in the Introduction and one sentence in the "Model and Formalism" section to reinforce the novelty of our manuscript.

4) We have fixed the typos in the caption of Fig. 2.

5) We have fixed the measurement units in the caption of Fig3.

6) We have added a new paragraph about Majorana Polarization at the end of the section: RESULTS.

7) We have added a new paragraph about experimental realization at the end of the section conclusions and outlooks.

8) We have added a new sentence in the Appendix C explaining in more detail the spectra depicted in Fig. 5.

---

## Editorial Decision

published